# Rugby Health and Well-Being Study: protocol for a UK-wide survey with health data cross-validation

Nirmala Kanthi Panagodage Perera,[1,2] Maja R Radojčić [1,2] Stephanie R Filbay [1,2] Steffan A Griffin,[3,4] Lucy Gates,[1,5] Andrew Murray,[4,6] Roger Hawkes,[7] Nigel K Arden[1,2]

¹Nuffield Department of Orthopaedics, Rheumatology and Musculoskeletal Sciences, University of Oxford, Oxford, UK
²Centre for Sport, Exercise and Osteoarthritis Research Versus Arthritis, University of Oxford, Oxford, UK
³Rugby Football Union, Twickenham, UK
⁴Centre for Sport and Exercise, University of Edinburgh, Edinburgh, UK
⁵Centre for Sport, Exercise and Osteoarthritis Research Versus Arthritis, University of Southampton, Southampton, UK
⁶Scottish Rugby Union, Murrayfield, UK
⁷British Association of Sport and Exercise Medicine, Doncaster, UK

**Correspondence to**
Dr Maja R Radojčić;
maja.radojcic@ndorms.ox.ac.uk

## ABSTRACT

**Introduction** Rugby football (Union and League) provides physical activity (PA) with related physical and mental health benefits. However, as a collision sport, rugby research and media coverage predominantly focus on injuries in elite players while the overall impact on health and well-being remains unclear. This study aims to provide a greater understanding of the risks and benefits of rugby participation in a diverse sample of men and women, current and former rugby Union and League players from recreational to the elite level of play. We will explore: (1) joint-specific injuries and concussion; (2) joint pain and osteoarthritis (OA); (3) medical and mental health conditions; (4) PA and sedentary behaviour and (5) well-being (quality of life, flourishing and resilience).

**Methods and analysis** The Rugby Health and Well-being Study is designed in two phases: (1) a UK-wide cross-sectional survey and (2) cross-validation using health register data from Scotland. Participants will be at least 16 years old, current or former rugby players who have played rugby for at least one season. We will report standardised, level of play-, sex- and age-stratified prevalence of joint injury, concussion, medical conditions and PA. We will describe injury/concussion prevention expectations and protective equipment use. Rugby-related factors associated with injury, pain, OA, PA, health and well-being will be explored in regression models. We will compare joint pain intensity and duration, elements of pain perception and well-being between recreational and elite players and further investigate these associations in regression models while controlling for confounding variables. In the second phase, we will validate self-reported with health register data, and provide further information on healthcare use.

**Ethics and dissemination**
The Yorkshire and the Humber—Leeds East Research Ethics Committee (REC reference: 19/HY/0377) has approved this study (IRAS project ID 269424). The results will be disseminated through scientific publications, conferences and social media.

## INTRODUCTION

Health is 'a state of complete physical, mental and social well-being and not merely the absence of disease or infirmity'[1] as defined by WHO. Growing evidence suggests that regular

---

### Strengths and limitations of this study

► A large nation-wide sample of current and former rugby participants of different playing standards and ages.
► A comprehensive data collection about injury experience and pain perception, physical activity and quality of life which will contribute to evidence-informed strategies to promote player welfare.
► The study will inform recommendations for the design of future prospective studies on health and well-being in current and former rugby players.
► Limitations include cross-sectional design, the potential for recall bias of self-reported data and unavailability of health register data for all participants for the validation phase.

---

physical activity (PA) can provide longevity, physical and mental health benefits regardless of age, sex, geographical location and socioeconomic backgrounds, and provide economic benefits for communities[2–4] and improve well-being.[5 6] According to the Compendium of Physical Activities criteria,[7] rugby football (referred to as rugby herein) can be a source of moderate-to-vigorous PA.[8 9] As a collision sport, there are risks of musculoskeletal injury and concussion[10–15] with acute injuries being the focus of the majority of research[16] and media coverage. The overall incidence of musculoskeletal injuries ranged from 19.6 for elite women to 62 and 148 injuries per 1 000 match-hours for amateur and elite men players, respectively.[17–19] Among these, injuries of the lower and upper limb joints, and neck have been the most commonly reported.[14] Additionally, the overall incidence of reported concussion ranged from 0.55 for professional women to 2.08 and 4.73 per 1 000 match-hours for amateur and professional men levels, respectively.[10] These injuries have been associated with adverse long-term health outcomes

like increased risk of chronic pain, post-traumatic osteoarthritis (OA)[20 21] and joint replacement,[22] neurocognitive deficits[23] and decreased neuropsychological performance[24] and impaired quality of life.[20 25–27] However, on top of the physical benefits of sport participation,[28–30] team sport-based psychosocial benefits include resilience, self-efficacy, enhanced coping, positivity and sense of belonging.[31–34] The overview of sport participation and medium/long-term health outcomes has been synthesised in football, tennis, golf[30] and cycling[35] but not in rugby.[16] Thus, there is a pressing need to evaluate the health and well-being benefits, as well as the risks involved in playing rugby.

The priority of global rugby governing bodies is player welfare and making rugby safe and enjoyable for all. Women and men throughout the world play rugby in two distinct codes: Union and League. There are many similarities between the codes and distinct differences like player morphology, tackle technique, a demographic and socioeconomic background that can influence overall benefit–risk relation. The rugby-related risk in different populations can be prevented by strategies that deal not only with prevention of injury occurrence but also with slowing disease development and reducing disability. The first step toward developing different prevention strategies is to define the magnitude, scope and characteristics of the problem through the systematic collection of data.[36]

Thus, the Rugby Health and Well-being Study aims to provide a comprehensive overview and a greater understanding of the potential risks and benefits of rugby participation. We will include current and former, recreational and elite rugby Union and League players, men and women from a variety of cultural backgrounds in UK to explore: (1) joint-specific injuries and concussion, (2) joint pain and OA, (3) medical and mental health conditions, (3) PA and sedentary behaviour and (5) well-being, that is, quality of life, flourishing and resilience.

## METHODS AND ANALYSIS
### Study design and participants
The Rugby Health and Well-being Study is designed as a cross-sectional study in two phases: phase 1—an online survey and phase 2—data cross-validation from health register (figure 1). The study is informed by the Cricket Health and Well-being Study. That study performed qualitative research work, that is, interviews and patient and public involvement (PPI) within the Centre for Sport, Exercise and Osteoarthritis Research Versus Arthritis, in the process of defining the content and wording of questions.[37 38] This study incorporates the valuable feedback on the Cricket Health and Well-being Study, and also adds several additional sections of interest, cross-validation phase and adaptations specific to rugby.

Inclusion criteria for this study are people aged 16 years or older, who are current or former rugby players having played rugby for at least one season. We aim to recruit

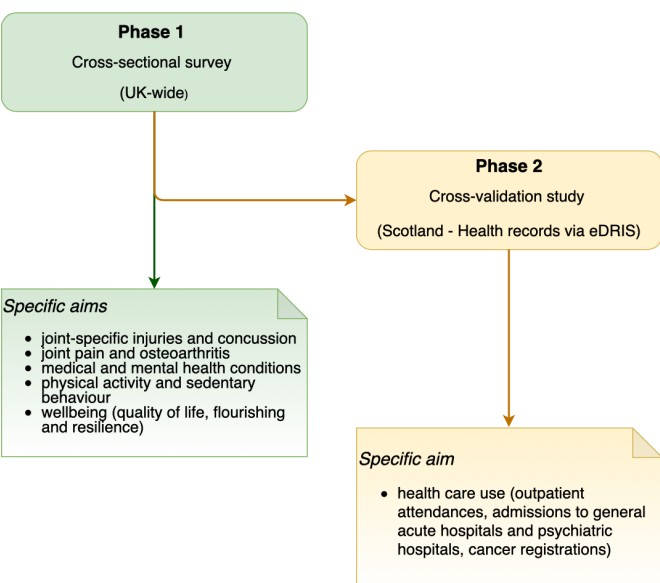

**Figure 1** Study design and aims. eDRIS, electronic data research and innovation service.

a diverse sample including recreational players, women and youth from a variety of cultural backgrounds as these groups are under-represented in existing rugby research.

### Sample size
The study sample size is calculated using the equation for surveys[39]:

$$\text{Sample size} = \frac{\frac{z^2 \times p(1-p)}{e^2}}{1 + \frac{z^2 \times p(1-p)}{e^2 N}}$$

where N: population size, p: proportion (standard for surveys 50%), e: margin of error (5%) and z: z-score (1.96 for 95% CI).

According to the equation and given constants, the sample size depends only on the population size (N). In 2016, there were 3 200 000 registered men and 27 500 women playing rugby in England.[40] For these populations, the required sample size is 385 men and 379 women. Given this slight difference, we assume equal-size groups, meaning that we need to recruit 770 current players for England. We expect similar numbers of former and current players to complete the survey such that we will need to recruit a total of 1540 players. To optimise the study to be representative of the UK, we considered that the population size of England is 80% of the UK population and extrapolated the same percentage to rugby players. Thus, we will aim to recruit an overall sample size of 1 925 players across the UK. Finally, based on a response rate (10%) in the Cricket Health and Well-being Study,[41] we need to contact 19 250 current and former players to achieve the intended sample size.

## Recruitment

We will employ several recruitment strategies to reach the intended sample size and diversity while ensuring stratified sampling per current and former players followed by simple random sampling. First, to recruit current rugby players, we will send emails to clubs registered with stakeholders asking to forward the recruitment letters to their players. Also, research team members will attend local clubs to provide further information and encourage participation. To reach former players as well, we will contact alumni organisations with established links to rugby in a similar manner asking to forward recruitment letters to their members. Further, in previous studies in sports conducted by our group, participants have been asked and consented to permission for future contact. We will contact these people by email. Additionally, there are online tools available (www.callforparticipant.com) where people have registered to participate in research studies. We will post the project details on this website to alert about our study. In all occasions, the email will provide information about the study, the Participant Information Sheet and a link to the survey.

Additionally, we will use standard advertisements and social media to reach a wider population. We will provide information about the study and ongoing recruitment through advertisements in newspapers, magazines, flyers, posters, information sheets, notices, internet postings, public engagement activities by our group and traditional media appearances such as radio. Additionally, a press release will be issued through Centre for Sport, Exercise and Osteoarthritis Research Versus Arthritis to alert the public about the study and to recruit participants. Furthermore, the research team will use social media platforms such as Twitter to share the Participant Information Sheet and a link to the survey. The study will also be promoted via podcasts, blog posts and newsletters.

## Phase 1

All participants will complete the online survey—Phase 1. The survey link is specific per email address ensuring that each participant completes the survey only once. The online survey will be administered via electronic data capture software ALEA (FormsVision, Abcoude, the Netherlands) through the University of Southampton. It will start in 2021 and be active for 6 months. The survey completion requires a time commitment of 20–45 min owing to branching logic.

The survey is organised in several sections according to the study aims. First, participants will report their demographic data such as age, height, weight, ethnicity, education level and employment status, as well as smoking, recreational drug and alcohol use.

### Joint-specific injury and concussion

We will collect data on playing status, rugby participation history, age and reason/s for retirement, joint injury and concussion history, protective equipment use, injury/ concussion risk perception, prevention expectations and knowledge.[42 43]

### Joint pain and osteoarthritis

Participants will report their current pain, as well as duration-based severe joint pain, that is, pain on most days per month in the previous year. They will be asked to identify all affected joints including laterality on a predetermined list. The joint pain parameters of interest include intensity, chronicity, occurrence and the effect on daily activities. Additionally, this part includes selected questions from the Pain-related Cognitive Processes Questionnaire[44] where participants are asked to report cognitive, behavioural and emotional aspects when experiencing pain. We will also collect data on the history of OA and orthopaedic surgery.

### Medical and mental health conditions

We will gather data about current and previous medical and mental health conditions diagnosed by a health professional,[20 45] stress, sleep and well-being,[43] effects of health problems on rugby participation[46 47] and access to healthcare.

### Physical activity and sedentary behaviour

This section aims to collect data on types of other sports played, PA and sedentary behaviour over the past 7 days using International Physical Activity Questionnaires – Short Form questionnaire[48] and perception of current PA levels.

### Well-being: quality of life, flourishing and resilience

The last section of the survey focuses on the quality of life using the Short Form Health Survey 12,[49] enjoyment and physical/psychological benefits of rugby, resilience (European Social Survey)[50] and flourishing (Flourishing Scale).[51]

### Comparison of population-based cohorts

We will use data from two representative population-based cohorts in England as a comparison to calculate standardised morbidity ratios (SMRs) of medical conditions reported in our study. For this purpose, participants of the Rugby Health and Well-being Study will be age- and sex-matched to participants from the English Longitudinal Study of Ageing and the Health Survey for England, 2015 data.[52]

## Phase 2

Phase 1 participants from Scotland whose consent to share their full name and date of birth to enable matching with their Community Health Index (CHI) number will be included in Phase 2. This phase is aimed to validate self-reported health data from Phase 1 and gather additional information on healthcare use. It is restricted to Scottish residents due to the availability of electronic health register via electronic Data Research and Innovation Service (eDRIS) (https://www.isdscotland.org/Products-and-Services/EDRIS). Through eDRIS research

support, prospectively collected comprehensive inpatient and outpatient electronic medical records from 1981 till date is available for all Scottish residents. The aim of eDRIS is to support research use of administrative datasets by providing a single-entry point and end-to-end support to researchers. The high-quality eDRIS data provide national coverage across Scotland and the ability to link data to allow patient-based analysis. The International Classification of Diseases 9/10 codes are used in these electronic medical records to identify conditions resulting in hospitalisation. After matching survey participants to their CHI number, individual-level data on outpatient attendances (SMR00), admissions to general acute hospitals (SMR01), psychiatric hospitals (SMR04) and cancer registrations (SMR06) will be obtained. These will be used for providing agreement between health register data and Phase 1 medical and mental health data and reporting additional information if not included in Phase 1.

## Statistical analysis

We will provide descriptive statistics for the participants in Phase 1 and Phase 2 and compare the rugby participants with a population-based comparison cohort for the result generalisation. We will report on the prevalence per 1000 participants of self-reported joint-specific injuries, concussion, joint-specific pain, OA and medical and mental health conditions. These will be standardised to the population but also stratified by age, sex and playing position. Further, we will calculate SMRs for medical and mental health conditions, OA, and joint replacement for rugby players when age- and sex-matched to participants from the population-based comparison cohort.

The explorative analyses will aim to determine the relationship between rugby-related factors and joint-specific injuries, joint-specific pain and OA for women and men, current and former rugby players of all standards of play. This analysis will determine factors from training and playing history that are most likely to be associated with an increased risk of OA. We will use multivariable regression models to assess the interactions and to control the associations for the influence of relevant confounding variables. Injury prevention expectations and risk perception will be compared between elite and recreational standards of play using a *t*-test or a Mann-Whitney *U* test. We will describe protective equipment use. We will investigate pain parameters and elements of pain perception and compare these between recreational and elite players using a *t*-test. The relationship between elements of pain perception with pain intensity and duration while controlling for rugby-related factors will be assessed in linear and logistic regression models. PA and sedentary behaviour in current and former rugby players will be analysed using descriptive statistics and stratified by age and sex. Binary logistic regression analyses will be performed to determine the odds of meeting UK PA guidelines compared with a representative population-based comparison cohort and exploring

factors associated with inactivity in former rugby players. The well-being of current and former rugby players will be explored using descriptive analyses. Factors related to quality of life, flourishing and resilience will be determined in multivariable regression models.

In the validation phase, we will provide a percentage of agreement between survey and health register data and descriptive statistics on the healthcare use not captured by Phase 1.

## Strengths and limitations

In this study, we will include a diverse nation-wide sample of rugby participants —men and women, current and former rugby Union and League players from recreational to the elite level of play. It will be a comprehensive data collection about injury and pain experience, PA and quality of life that will contribute to evidence-informed strategies to promote player welfare and enhance the positive aspects of rugby. In a subsample, we will cross-validate self-reported data with electronic health records. Results of this study will provide recommendations for the design of future prospective studies.

Nevertheless, the study has some limitations. First, it is a cross-sectional design with retrospective questions. We will investigate areas of interest in a single time point without considering time-dependent relations. However, this study will inform the design of future prospective rugby studies. Second, survey data are self-reported, and participants are asked about past events often subjected to recall bias. We have included several steps to minimise the recall bias. We ask about major injury that is, injury leading to more than 4 weeks of reduced participation in exercise, training or sport, the joint pain lasting most days per month and medical and mental health conditions diagnosed by healthcare professionals only. Further, the accuracy of retrospective self-reported injury compared to prospective reports has been shown in recreational Australian Football League players. Whereby, there was perfect agreement between a retrospective injury survey and prospective injury records over 12 months.[53] Also, 79% of the study participants accurately recalled the number of injuries, and 61% recalled the number, anatomical location and diagnoses of these injuries. Overall, the impact of recall bias is toward under-reporting injuries and concussion.[53 54] Yet, under-reporting other than recall bias can be an issue in current players. Our Participant Information Sheet includes a clarification that coaches/managers will not have access to the survey. Overall, the under-reporting can potentially shift our results toward zero. Thirdly, our Phase 2 has some limitations, as well. While it is an objective way to validate self-reported health data and deal with under-reporting, it is available for a subsample only. We cannot control the inclusion of the participants in the subsample, but we will report and discuss how the subsample represents the whole study sample. Also, we are limited by diagnostic codes and unable to explore all the health conditions. Finally, owing to the time commitment required for completing the survey, we

prioritise the areas of interest over additional areas that potentially could confound some of the associations of interest, such as family history.

## Patient and public involvement

Centre for Sport, Exercise and Osteoarthritis Research Versus Arthritis has a PPI representative fully engaged in the research work providing suggestions and directions to explore, as well as feedback on the findings. We included, in this study, the PPI suggestions on diversity and interest in recreational players. In designing the survey, we involved men and women current and former rugby players and coaches, and international rugby representatives. They ensured that the survey is relevant and specific for rugby players to collect accurate and sport-specific self-reported data.

## ETHICS AND DISSEMINATION

The Yorkshire and the Humber—Leeds East Research Ethics Committee (REC reference: 19/HY/0377) has approved the study (IRAS project ID 269424) in accordance to the Declaration of Helsinki of the World Medical Association. Written informed consent will be obtained from all participants. For young people aged 16 years or older, the UK law dictates that they can provide consent for medical treatment unless they are not able to make the decision. Following the same logic, individuals 16 years or older can consent to participate in this study.

We will use a multimodal approach for dissemination and knowledge translation. Following the publication policy of Versus Arthritis, results will be published in open access peer-reviewed journals to enhance dissemination. In addition to scientific conference presentations, we will employ multimedia resources like infographics, animations, podcasts and blogs to subsequently disseminate and communicate results to the study participants, general public and stakeholders via social media platforms and mass media.

**Contributors** NKPP produced the first draft of the manuscript; MRR, SRF, SAG, LG, AM, RH and NKA provided critical comments on the manuscript. All authors were involved in designing the study and approved the final draft for submission.

**Funding** This work was supported by Versus Arthritis grant number 21595—Centre for Sport, Exercise and Osteoarthritis Research Versus Arthritis.

**Competing interests** NKPP, MRR, SRF, LG, RH and NKA declare no competing interests. SAG works as a Sports Medicine Training Fellow for the Rugby Football Union and receives remuneration for clinical work in the sport. AM receives remuneration from Scottish Rugby Union for clinical work.

**Patient and public involvement** Patients and/or the public were involved in the design, or conduct, or reporting, or dissemination plans of this research. Refer to the Methods section for further details.

**Patient consent for publication** Not required.

**Provenance and peer review** Not commissioned; externally peer reviewed.

**ORCID iDs**
Maja R Radojčić http://orcid.org/0000-0001-8797-6142
Stephanie R Filbay http://orcid.org/0000-0002-9624-0791

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
