## [Reviewer comments · BMJ Open]

ARTICLE DETAILS

TITLE (PROVISIONAL)	The Rugby Health and Wellbeing Study: Protocol for a UK-wide Survey with Health Data Cross-validation
AUTHORS	Panagodage Perera, Nirmala; Radojčić, Maja; Filbay, Stephanie; Griffin, Steffan; Gates, Lucy; Murray, Andrew; Hawkes, Roger; Arden, Nigel

VERSION 1 – REVIEW

REVIEWER	Juan José Ramos-Álvarez Complutense University Madrid (Spain)
REVIEW RETURNED	18-Oct-2020

GENERAL COMMENTS	The authors explain the strategies used to recruit the sample. It is suggested that they define the type of sampling performed. They should provide bibliography for the formula used to determine sample size. It would be interesting if the survey asked about the time of absence from the competition after a concussion, since that point is currently under debate.
---

REVIEWER	Moore, IS Cardiff Metropolitan University
REVIEW RETURNED	09-Nov-2020

GENERAL COMMENTS	The study aims to provide a comprehensive evaluation of the impact rugby football has on several health-related outcomes, both physical and mental. The authors are undertaking a much needed study and have pulled together a large survey to address the study's aims. Whilst the study is retrospective in nature and will likely suffer from recall bias, which will possibly be exacerbated in the former players cohort (relative to the current players), Phase 2 will be a useful cross-validation step that goes beyond what many have done previously. In general, I have no major concerns. I do have some comments relating to the structure of the argument in the introduction, in terms of its relatedness to rugby and the use of an England only database for the sample size calculation. Abstract Line 28: suggest rephrasing to 'predominantly focuses' Line 44: word missing 'controlling for confounding X.' Introduction Line 71: the current phrasing almost sounds like concussion is not an injury per se. To support the focus on concussion in the study, it would be useful to re-word this slightly to say '...risks of injury, with concussion being the most common acute injury' or
---

	something similar. Perhaps include epidemiology data to further support the point. Line 72 – 77: There are rugby specific injury risks noted, but then general risks and benefits outlined. I think this could be tailored more towards rugby. For example, you could note common injuries in rugby and then outline clearly potential long-term adverse effects for these injuries. From the list provided, I suspect that you have most of these covered, but it currently reads as a bit too generic for ‘sports injuries’ some of which may not be a huge issue for rugby. Similarly for the psychosocial, you could draw upon the fact rugby is a team sport and therefore bring in team-sport based benefits. Line 94: given the aims are quite specific about joint injuries, I think this needs justifying more strongly in the earlier narrative. This could be woven into my first comment on the introduction about specific rugby injuries. Methods: Lines 116 – 125: Given that England has the largest population within the UK, the sample size I suspect is an overestimation. Indeed, it would also lead to an over-representation of Scotland, Wales and Northern Ireland. This would need mitigating somehow either by following the suggestion provided or if that approach can’t be undertaken, finding another solution. Suggestion: It would be a stronger approach to get the registered numbers for the UK rather than extrapolating from England. If this are available, please include them and revise this paragraph accordingly. Line 182: Why just England based comparisons rather than UK? Can UK ones be used instead? Please name the two representative population-based cohorts (and hyperlinked if applicable). Line 215-232: Are you considering sex or gender, and in which analysis will this be included? A good overview of elite to recreational, former to current, is provided. However, there is no mention of sex and/or gender. Is this a confounding variable only? Given the different levels of exposure to coaching/support staff that occurs in men and women’s rugby it may be useful for injury prevention and or pain analysis to include assessing the potential for a sex and/or gender difference.
--	--

VERSION 1 – AUTHOR RESPONSE

Point-by-point reply to reviewers’ comments

REVIEWER #1

1. The authors explain the strategies used to recruit the sample. It is suggested that they define the type of sampling performed.

Reply: *We thank the reviewer for the great suggestion. Here is the change we made.*

We will employ several recruitment strategies to reach the intended sample size and diversity while ensuring stratified sampling per current and former players followed by simple random sampling.

2. They should provide bibliography for the formula used to determine sample size.

Reply: We thank the reviewer for pointing to the lack of the reference. We included reference 40 (Page 6, Line 121) Daniel WW. *Biostatistics: A Foundation for Analysis in the Health Sciences, Textbook + Student Solutions Manual: John Wiley & Sons Inc, New York. 2005.*

3. It would be interesting if the survey asked about the time of absence from the competition after a concussion, since that point is currently under debate.

Reply: We are thankful for this comment, and we agree that it is an important question to address. We did not specify each question in the manuscript, but Page 7, Lines 176-179 include concussion history, and in our questionnaire, we specifically ask for the number of concussions and how many days it took the participants to return to full participation in exercise, training or sport after the most recent concussion.

REVIEWER #2

1. The study aims to provide a comprehensive evaluation of the impact rugby football has on several health-related outcomes, both physical and mental. The authors are undertaking a much needed study and have pulled together a large survey to address the study's aims. Whilst the study is retrospective in nature and will likely suffer from recall bias, which will possibly be exacerbated in the former players cohort (relative to the current players), Phase 2 will be a useful cross-validation step that goes beyond what many have done previously. In general, I have no major concerns. I do have some comments relating to the structure of the argument in the introduction, in terms of its relatedness to rugby and the use of an England only database for the sample size calculation.

Reply: We are delighted with the reviewer's compliments to our study and thankful for the feedback and directions for further improvement. We addressed the comments below and hope the revisions and responses are satisfactory.

Abstract

2. Line 28: suggest rephrasing to 'predominantly focuses'

Reply: We made the change as suggested.

Page 2, Line 28

However, as a collision sport, rugby research and media coverage predominantly focus on injuries in elite players while the overall impact on health and wellbeing remains unclear.

3. Line 44: word missing 'controlling for confounding X.'

Reply: We included the word variables.

Page 2, Line 44

We will compare joint pain intensity and duration, elements of pain perception and wellbeing between recreational and elite players and further investigate these associations in regression models while controlling for confounding variables.

Introduction

4. Line 71: the current phrasing almost sounds like concussion is not an injury per se. To support the focus on concussion in the study, it would be useful to re-word this slightly to say '...risks of injury, with concussion being the most common acute injury' or something similar. Perhaps include epidemiology data to further support the point.

Reply: We specified injury to be related to the musculoskeletal system and distinguished from the concussion. As indicated, we supported these with appropriate available statistics.

Page 4, Lines 71-78

As a collision sport, there are risks of musculoskeletal injury and concussion¹⁰⁻¹⁵ with acute injuries being the focus of the majority of research¹⁶ and media coverage. The overall incidence of musculoskeletal injuries ranged from 19.6 for elite women to 62 and 148 injuries per 1,000 match-hours for amateur and elite men players, respectively.¹⁷⁻¹⁹ Among these, injuries of the lower and upper limb joints, and neck have been the most commonly reported.²⁰ Additionally, the overall incidence of reported concussion ranged from 0.55 at professional women to 2.08 and 4.73 per 1,000 match-hours at amateur and professional men level, respectively.¹⁰

5. Line 72 – 77: There are rugby specific injury risks noted, but then general risks and benefits outlined. I think this could be tailored more towards rugby. For example, you could note common injuries in

rugby and then outline clearly potential long-term adverse effects for these injuries. From the list provided, I suspect that you have most of these covered, but it currently reads as a bit too generic for 'sports injuries' some of which may not be a huge issue for rugby. Similarly for the psychosocial, you could draw upon the fact rugby is a team sport and therefore bring in team-sport based benefits.

Reply: *We adjusted as suggested. The most common injuries in rugby are included in the previous comment, and we continued that these have been generally associated with long-term outcomes, and stressed team-sport based benefits as well.*

Page 4, Lines 78-83

These injuries have been associated with adverse long-term health outcomes like increased risk of chronic pain, post-traumatic osteoarthritis (OA)^{21 22} and joint replacement,²³ neurocognitive deficits²⁴ and decreased neuropsychological performance²⁵ and impaired quality of life.^{21 26-28} However, on top of the physical benefits of sport participation,²⁹⁻³¹ team-sport based psychosocial benefits include resilience, self-efficacy, enhanced coping, positivity and sense of belonging.³²⁻³⁵

6. Line 94: given the aims are quite specific about joint injuries, I think this needs justifying more strongly in the earlier narrative. This could be woven into my first comment on the introduction about specific rugby injuries.

Reply: *We hope that replies to the previous two comments effectively addressed this point.*

Methods:

7. Lines 116 – 125: Given that England has the largest population within the UK, the sample size I suspect is an overestimation. Indeed, it would also lead to an over-representation of Scotland, Wales and Northern Ireland. This would need mitigating somehow either by following the suggestion provided or if that approach can't be undertaken, finding another solution. Suggestion: It would be a stronger approach to get the registered numbers for the UK rather than extrapolating from England. If this are available, please include them and revise this paragraph accordingly.

Reply: *We thank the reviewer very much for this comment and appreciate drawing our attention on the overestimation of the Scotland, Wales and Northern Ireland in our sample size calculation. We were not able to find registered numbers of Rugby Union and League players for the UK, only England. We kept the same calculation for England but adjusted the overall sample size taking England population being 80% of the UK population and extrapolating the percentage to rugby players.*

Page 6, Lines 131-136

To optimise the study to be representative of the UK, we considered that the population size of England is 80% of the UK population and extrapolated the same percentage to rugby players. Thus, we will aim to recruit an overall sample size of 1,925 players across the UK. Finally, based on a response rate (10%) in the Cricket Health and Wellbeing Study,⁴² we need to contact 19,250 current and former players to achieve the intended sample size.

8. Line 182: Why just England based comparisons rather than UK? Can UK ones be used instead? Please name the two representative population-based cohorts (and hyperlinked if applicable).

Reply: *We are thankful for this comment. Unfortunately, we do not have open access population-based cohorts in Wales, Scotland or Northern Ireland or the UK overall. However, as discussed in the previous comment, England is 80% of the UK, thus, the representative cohort of England is very likely to be representative of the UK. If in meanwhile, we get access to any other cohort, we will be happy to use data for relevant comparisons in our study. The names of the cohorts we intend to use are the English Longitudinal Study of Ageing (ELSA) and the Health Survey for England (HSE) (Page 8, Lines 205-206).*

9. Line 215-232: Are you considering sex or gender, and in which analysis will this be included? A good overview of elite to recreational, former to current, is provided. However, there is no mention of sex and/or gender. Is this a confounding variable only? Given the different levels of exposure to coaching/support staff that occurs in men and women's rugby it may be useful for injury prevention and or pain analysis to include assessing the potential for a sex and/or gender difference.

Reply: *We thank the reviewer very much for pointing to oversight considering sex in our analyses and confusion it can cause. We will use sex as a confounding variable and for stratification depending on the research question, analysis and satisfaction of epidemiological principles for stratification. We did make it clear in Abstract (Line 38), however, we acknowledged that section Statistical analysis needed improvement on this matter, and we included several clarifications.*

Page 9

Lines 229-230

These will be standardised to the population but also stratified by age, sex and playing position.

Page 10

Lines 234-236

The explorative analyses will aim to determine the relationship between rugby-related factors and joint-specific injuries, joint-specific pain and osteoarthritis for women and men, current and former rugby players of all standards of play.

Lines 245-247

Physical activity and sedentary behaviour in current and former rugby players will be analysed using descriptive statistics and stratified by age and sex.

VERSION 2 – REVIEW

REVIEWER	Juan José Ramos Álvarez Universidad Complutense de Madrid
REVIEW RETURNED	27-Dec-2020
GENERAL COMMENTS	No comment. The authors have corrected the suggestions regarding sample size.